# New Smart Bioactive and Biomimetic Chitosan-Based Hydrogels for Wounds Care Management

**DOI:** 10.3390/pharmaceutics15030975

**Published:** 2023-03-17

**Authors:** Simona-Maria Tatarusanu, Alexandru Sava, Bianca-Stefania Profire, Tudor Pinteala, Alexandra Jitareanu, Andreea-Teodora Iacob, Florentina Lupascu, Natalia Simionescu, Irina Rosca, Lenuta Profire

**Affiliations:** 1Department of Pharmaceutical Chemistry, Faculty of Pharmacy, University of Medicine and Pharmacy “Grigore T. Popa” of Iasi, 16 Universitatii Street, 700115 Iași, Romania; 2Research & Development Department, Antibiotice Company, 1 ValeaLupului Street, 707410 Iasi, Romania; 3Department of Analytical Chemistry, Faculty of Pharmacy, University of Medicine and Pharmacy “Grigore T. Popa” of Iasi, 16 Universitatii Street, 700115 Iași, Romania; 4Department of Internal Medicine, Faculty of Medicine, University of Medicine and Pharmacy “Grigore T. Popa” of Iasi, 16 Universitatii Street, 700115 Iași, Romania; 5Department of Orthopedics and Traumatology, Faculty of Medicine, University of Medicine and Pharmacy “Grigore T. Popa” of Iasi, 16 Universitatii Street, 700115 Iași, Romania; 6Department of Toxicology, Faculty of Pharmacy, University of Medicine and Pharmacy “Grigore T. Popa” of Iasi, 16 Universitatii Street, 700115 Iași, Romania; 7Centre of Advanced Research in Bionanoconjugates and Biopolymers, “PetruPoni” Institute of Macromolecular Chemistry, 41A GrigoreGhica-Voda Alley, 700487 Iasi, Romania

**Keywords:** smart hydrogels, oxidized chitosan, oxidized hyaluronic acid, nontoxic crosslinkers, APIs, wound healing, self healing, self adapting

## Abstract

Wound management represents a continuous challenge for health systems worldwide, considering the growing incidence of wound-related comorbidities, such as diabetes, high blood pressure, obesity, and autoimmune diseases. In this context, hydrogels are considered viable options since they mimic the skin structure and promote autolysis and growth factor synthesis. Unfortunately, hydrogels are associated with several drawbacks, such as low mechanical strength and the potential toxicity of byproducts released after crosslinking reactions. To overcome these aspects, in this study new smart chitosan (CS)-based hydrogels were developed, using oxidized chitosan (oxCS) and hyaluronic acid (oxHA) as nontoxic crosslinkers. Three active product ingredients (APIs) (fusidic acid, allantoin, and coenzyme Q10), with proven biological effects, were considered for inclusion in the 3D polymer matrix. Therefore, six API-CS-oxCS/oxHA hydrogels were obtained. The presence of dynamic imino bonds in the hydrogels’ structure, which supports their self-healing and self-adapting properties, was confirmed by spectral methods. The hydrogels were characterized by SEM, swelling degree, pH, and the internal organization of the 3D matrix was studied by rheological behavior. Moreover, the cytotoxicity degree and the antimicrobial effects were also investigated. In conclusion, the developed API-CS-oxCS/oxHA hydrogels have real potential as smart materials in wound management, based on their self-healing and self-adapting properties, as well as on the benefits of APIs.

## 1. Introduction

The management of acute and chronic wounds represents a challenge and a continuous concern of the health system worldwide. The prevalence of all types of injuries is increasing every year due to an aging population and the growing incidence of wound-related comorbidities, such as diabetes, high blood pressure, obesity, autoimmune diseases, and peripheral vascular dysfunctions [1,2,3]. Among the patients hospitalized for acute conditions, 25–50% present or develop wounds during hospitalization, with high risks of infection and chronicity [4,5].

Nowadays, various products are available in the wound care market, such as dressings, creams, ointments, gels, sprays, and powders [6]. After performing the debridement, choosing the appropriate dressing is the next step in wound management, and it is crucial for the final outcome of the healing process [7,8]. It requires a balance between benefits, safety for the patient, and cost effectiveness [9]. The dressing should also be adapted to patient preferences and clinical conditions in order to ensure compliance and treatment success [10]. Topical treatment for wound management requires the fulfillment of several conditions to ensure and maintain a favorable environment for the entire healing process [11,12].

The ideal wound dressing should be biocompatible and biodegradable, and it should fit perfectly to the wound’s shape, though not too adherent to the wound surface, provide protection against mechanical and thermal stress, prevent microbial contamination, and optimize the moisture in the affected tissue in order to promote effective, rapid, and painless healing [13,14].

Biopolymers are a good substitute for traditional wound healing agents due to their benefits such as biocompatibility, biodegradability, bioactivity, bioresorptivity, no-toxicity, enhanced antithrombin activity, and anticoagulant activity, protection from mechanical stress, drying, infections, and radiation, and effective gelling and swelling ability [15,16]. So, various biopolymers-based dressings, such as creams, films, powders, sponges, electrospun fibers, and hydrogels were developed according to the requirements for the healing process [17,18].

In many aspects, hydrogels are considered ideal for wound care since they mimic the skin structure, promote growth factor synthesis, and the autolysis process [19]. Defined by their 3D networks, hydrogels are capable of encapsulating large amounts of water or biological fluids, which makes them suitable for the management of draining painful wounds, radiation wounds, minor burns, or dry wounds [20,21]. Due to their hydrophilic character, oxygen permeability, the ability of diffusion, cell adhesion, ability to incorporate and release a wide variety of therapeutic agents, and ease in topical application, hydrogels represent a very important alternative in wound treatment [20,22]. Along with the positive aspects, some drawbacks reduce the applicability of hydrogels in medical practice. These include poor mechanical stability (the viscosity decreases over time), high risk for microbial contamination (due to the high amount of water content), toxicity potential (due to unreacted molecules when crosslinking agents are used), degradation, and high variability in release profile of the incorporated active substances [20,23].

To overcome these disadvantages, a wide variety of strategies were proposed for hydrogel development. Different types of polymers (synthetic, semisynthetic, or naturals) and physical or chemical crosslinking methods were proposed in order to enhance hydrogel performance and extend their applications as tissue (bone and cartilage) engineering, cell therapeutics, wound healing, controlled drug release, biosensors, and medical devices [21,24]. If, in the past, the main role of hydrogels was to protect the damaged tissue, at present, extensive research is being conducted for “smart” materials that not only protect the wounds but have the ability to influence all stages of healing [25].

Crosslinking is mandatory in hydrogel development in order to achieve a stable-over-time 3D matrix. The traditionally used physical and chemical crosslinking methods have demonstrated over time some undesirable effects on polymer functionality (structure rigidity) or tissue cytotoxicity (inhibition of epidermal cell proliferation and adhesion) [26].

The dynamic crosslinking methods have gained more attention lately as studies suggest that using chemical interactions and versatile hydrogels may be developed [27]. These chemical interactions are capable of uncoupling and recoupling if the dynamic equilibrium of the reaction is achieved or exceeded [24,28]. For example, oxidized alginate, polyethylene glycol dibenzaldehyde, and oxidized polysaccharides (hyaluronic acid and chitosan) were investigated for their crosslinking capacity via Schiff bases linkage in order to develop innovative nontoxic self-healing hydrogels for different medical applications [27,29]. The presence of a Schiff bases linkage (dynamic imine bonds) allows hydrogels to modify their 3D matrix, so as to adapt perfectly to the shape and depth of a wound (self-adapting ability). In this way, intimate contact with the damaged tissue is ensured, the healing processes are accelerated, and the protection against harmful environmental factors is higher [28,30]. The self-healing capability allows hydrogels to reorganize their structure according to the environmental conditions to which they are subjected. As a result, the hydrogels will have superior stability over time and exposure to various stressors (pH changes, temperature variations, and mechanical shear), compared to hydrogels obtained by conventional crosslinking methods [31,32].

Chitosan (CS) is currently one of the most studied biopolymers, having a polysaccharide structure that gives it versatility for numerous practical applications. It is used in the formulation of solutions, suspensions, hydrogels, micro- and nanoparticles, nanofibers, and sponges [33,34,35]. Among the uses of CS in biomedicine are synthesis of artificial skin, surgical sutures, artificial blood vessels, controlled release of drugs, tumor inhibition, and acceleration of wound healing [26,36]. The properties that recommend CS for these uses are biocompatibility, biodegradability, nontoxicity, mucoadhesiveness, and the ability to modulate physiological processes and antimicrobial properties [35,37]. CS also accelerates the wound-healing process by stimulating the migration of inflammatory cells, macrophages, and fibroblasts [37]. In this way, the inflammatory phase is reduced and the proliferative phase starts earlier in the wound-healing process [38]. The in vitro and in vivobehavior of CS depends on its molecular weight, deacetylation degree (DDA), and viscosity [39]. In vitro wound healing studies concluded that the higher the DDA, the more activated fibroblasts were founded in the proliferative phase of healing [40]. In vivo studies on animals have proven superior effects of CS in the treatment of burn wounds compared to heparin, and superior effects of CS with high DDA and molecular weight compared to CS batches with small DDA (75%) and low molecular weight (˂100 kDa) [41,42]. The study of infected postburn wounds in mice treated with topical antibiotics embedded in a CS-based hydrogel showed a significant decrease in mortality from 90% to 14%, as well as an acceleration of the healing process [43].

In order to improve its physicochemical properties and biological effects, different functionalized CS derivatives (glycol-CS, tannic acid-CS, carboxymethyl/β-tricalcium phosphate-CS, quaternized CS-graft-polyaniline, CS-grafted-dyhidrocaffeic acid, and agarose-CS) were obtained, using various chemical pathways (alkylation, acylation, quaternization, thiolation, sulfation, graft copolymerization, etc) [33,39,44,45]. Moreover, by crosslinking functionalized CS derivatives using oxidized polymers, versatile hydrogels for a variety of medical purposes (cartilage repair, bone regeneration, and wound healing) were developed [46,47,48,49,50,51,52].

Hyaluronic acid (HA) is a key molecule in the medical, pharmaceutical, nutrition, and cosmetic fields. Medical studies have demonstrated that HA is involved in wound regeneration processes, cell proliferation and migration, and tissue hemodynamics [53]. The effects on the wound healing process depend on its molecular weight. HA with a low molecular weight (50–200 kDa) has been demonstrated to be responsible for extracellular matrix regenerations through the stimulation of proteoglycans and fibronectin synthesis by fibroblasts [54,55,56]. Also, cytokine motility, angiogenesis, inflammatory effects, and oxidative stress may be modulated using products with HA with low molecular weight in wound treatments [57,58].

The oxidation of polysaccharides, such as CS, HA, cellulose, pectin, and alginates, was reported to enhance their water solubility and backbone flexibility due to the opening of the glucopyranose ring [59]. Moreover, the Schiff bases obtained through the reaction of carbonyl groups introduced in their structure proved superior antifungal and antibacterial properties [60]. These aspects show a promising perspective for using oxCS and oxHA as crosslinkers in the design of innovative wound-healing hydrogels based on bio-polymers.

The aim of this study was to develop new 3D bioactive and biomimetic CS-based hydrogels, with self healing and self adapting properties for wound care, using oxCS and oxHA as nontoxic crosslinkers. Three active product ingredients (APIs), fusidic acid, allantoin, and coenzyme Q10 (Figure 1), with proven biological effects, were considered for inclusion in the 3D polymer matrix.

Fusidic acid (FA) is a traditional antimicrobial agent recommended in the treatment of skin infections such as impetigo, infected wounds, folliculitis, abscesses, and erythrasma, being active both on aerobic and anaerobic germs [61,62]. This antibiotic has regained the interest of medical professionals as a result of the accelerated increase in antibiotic resistance used currently in topical infections. It acts on microbial metabolism as an inhibitor of protein synthesis at the level of the bacterial cell by disrupting the turnover of the elongate factor in ribosomes [63]. Topical products containing fusidic acid are available as ointments with 2% sodium fusidate, creams with 2% fusidic acid, and ophthalmic gels with 1% fusidic acid [62,63].

Allantoin (Ala) acts as a skin regeneration factor, contributes to collagen synthesis, accelerates the healing processes, and has a protective action against irritating factors, being included in the composition of topical products in concentrations of 0.5–2% [60,64,65,66].

The use of CoQ10 in topical treatments has proven beneficial by activating energy production in mitochondria and reducing the level of free radicals, resulting in antioxidant effects [67,68,69]. In the scientific literature, there are presented data that support CoQ10 exhibiting anti-inflammatory effects and favoring the wound healing processes [67]. Moreover, CoQ10 also showed significant antioxidant activity in vivo on malondialdehyde and superoxide dismutase levels, stimulating collagen synthesis (by scavenging collagenases), which promotes faster extracellular matrix recovery [68,69].

The novelty of our study is supported by the original design of the polymer matrix, CS-based hydrogels crosslinked with oxCS or oxHA, and having embedded FA, Ala, and CoQ10 as APIs, not being reported in the literature. The structure of the developed hydrogels, as well as the embedding of APIs, was proven using spectral methods (FTIR and NMR). The developed hydrogels were characterized by SEM, swelling degree, pH, and the internal organization of the 3D matrix was studied by rheological behavior. In addition, the cytotoxicity degree and the antimicrobial effects were also investigated. Based on their self-healing and self-adapting properties, as well as biological effects, these hydrogels have great potential to be used for a wide range of applications, including wound healing.

## 2. Materials and Methods

### 2.1. Materials

Chitosan with a medium molecular weight (200–300 kDa, DDA > 85%, viscosity of 200–800 cP) and hyaluronic acid sodium salt from *Streptococcus equi*, with low molecular weight (100–230 kDa) were pharmaceutical grade; lactic acid (99%), sodium periodate (99.8%), ethylenglicol anhydrous (98%) were analytical grade; allantoin (Ala), fusidic acid (FA) and coenzyme Q10 (CoQ10) were micronized pharmaceutical grade powders. All these chemicals were purchased from Sigma-Aldrich (Merck Group, Schnelldorf, Germany) and were used without any further purification. Dialysis tubing cellulose membrane with a molecular weight cut off of 14,000 Da was also purchased from Sigma-Aldrich. Human fibroblasts (HGF, CLS Cell Lines Service GmbH, Eppelheim, Germany), MEMα medium (Gibco, Thermo Fisher Scientific, Waltham, MA, USA), fetal bovine serum (FBS, Gibco, Thermo Fisher Scientific, Waltham, MA, USA), and 1% Penicillin-Streptomycin-Amphotericin B mixture (Lonza, Basel, Switzerland) were also used. Gram-negative (*Escherichia coli*ATCC 25922) and Gram-positive (*Staphylococcus aureus*ATCC 25923) bacterial strains aswell as pathogenic yeast (*Candida albicans*ATCC 90028) were provided by Mecconti, Poland.

### 2.2. Methods

#### 2.2.1. Preparation of CS-oxCS/oxHA and API-CS-oxCS/oxHA Hydrogels

Using the oxidized polymers (oxCS, oxHA) as crosslinking agents, which were obtained according to the method reported in the literature [46,48] (Appendix A), CS-based hydrogels were prepared. First, CS (1.0 g, 1.5 g, and 2.0 g) was hydrated in a sufficient amount of distilled water, at 50 °C for 15 min, under magnetic stirring (200 rpm), and then lactic acid (0.7 mL in 10 mL distilled water) was poured slowly and stirred again for other 10 min. Distilled water up to 100 g was added and stirring (200 rpm) and heating (50 °C) were continued for another 10 mi, when different CS solutions (1%, 1.5%, 2%, *w*/*w*) were obtained. Second, oxCS and oxHA solutions (1%, 1.5%, 2%, *w*/*w*) were prepared by mixing corresponding amounts of oxidized polymer with the appropriate quantity of distilled water and heated at 50 °C, under magnetic stirring (200 rpm). Finally, the oxidized polymer solution (oxCS/oxHA) was added gradually over the CS solution, in different ratios (Table 1), followed by stirring (300 rpm) until gelation occurred. Therefore, six CS-oxCS/oxHA hydrogels were prepared.

For the inclusion of APIs into the 3D polymer matrix, two CS-based hydrogels were selected: CS1.5-oxCS1.5 and CS1.0-oxHA2.0. The selection was based on the results of physicochemical tests, morphological characterization, and rheological behavior. The best morphological network was defined with the highest rate of structural recovery after exposure to very high shear stress (thixotropic test).

The selected APIs were inglobated directly to the CS-based hydrogels because they are slightly soluble in water.So, the bioactive CS-based hydrogels (API-CS-oxCS/oxHA) were prepared similarly to the CS-based hydrogels. Briefly, the corresponding APIs (FA, Ala, and, CoQ10, respectively) were dispersed by magnetic stirring (10 min, 300 rpm) in a sufficient amount of distilled water (10–15 mL), at room temperature (23 ± 2 °C) and then was added over the CS-oxCS/oxHA hydrogels and stirred again for an extra 10 min. As result, six API-CS-oxCS/oxHA hydrogels (FA-CS-oxCS, Ala-CS-oxCS, CoQ10-CS-oxCS, FA-CS-oxHA, Ala-CS-oxHA, and CoQ10-CS-oxHA) were prepared (Table 2).

#### 2.2.2. Physicochemical Characterization of CS-oxCS/oxHA and API-CS-oxCS/oxHA Hydrogels

The developed CS-based hydrogels (CS-oxCS/oxHA and API-CS-oxCS/oxHA) were characterized by a spectral analysis (FT-IR) in order to confirm the Schiff base formation (CS-oxCS/oxHA) and API inclusion (API-CS-oxCS/oxHA). Their morphology was highlighted using scanning electron microscopy (SEM), while the microstructure and the self-healing ability were evaluated by their rheological behavior. In addition, the swelling degree, pH, and 3D matrix appearances were also studied and correlated.

##### Macroscopic Aspect and Microscopic Analysis

The macroscopic aspect was evaluated by a visual observation of the hydrogels in natural light, while for the microscopic analysis, a Zeiss A Scope optical microscope (VIB, Gent, Belgium) with polarized light and objective ×40 was used.

##### Fourier-Transform Infrared Spectroscopy (FT-IR)

The FT-IR characterization of the CS-oxCS/oxHA and API-CS-oxCS/oxHA was carried out using the ABB-MB 3000 FT-IR MiracleTM Single Bounce ATR-crystal ZnSe system, in the wavelength range 500–4000 cm^−1^. Sixteen scans were performed at a resolution of 4 cm^−1^ for each determination. The obtained spectra were interpreted with Horizon MBTM FT-IR software version number-3.1.29.5.

##### Scanning Electron Microscopy (SEM)

The SEM images of the CS-oxCS/oxHA and API-CS-oxCS/oxHA hydrogels (lyophilized samples) were recorded using a HITACHI mass microscope (Nitech, Krefeld, Germany) at an acceleration voltage of 5–15 kV and under SE (secondary electron) detector. Secondary electrons are generated near the surface regions of the samples and carry information about the surface characteristics, being suitable to study the morphology and topography of a material by providing high-resolution images [70].

##### Swelling Degree Test

The swelling degree (SD) of CS-oxCS/oxHA and API-CS-oxCS/oxHA hydrogels was evaluated in the same conditions, using lyophilized samples. The samples (1 cm in diameter) were weighed before and after immersion in a phosphate buffer solution (PBS), pH = 7.4. The samples were taken off from the PBS every 1 h, lightly dabbed with filter paper to remove excess solution, and reweighed. The operation was repeated until the mass of the hydrated sample was constant. The SD (%) was calculated using the following equation:SD (%) = (M_t_ − M_0_)/M_0_ × 100(1)
where: M_0_ is the weight of the sample before immersion and M_t_ is the weight of the sample at a different time. The experiments were performed in triplicate.

##### Determination of pH

The pH of the hydrogels (CS-oxCS/oxHA and API-CS-oxCS/oxHA) was determined using a Melter Toledo pH-meter (Themo Fisher Scientific, Vienna, Austria), calibrated, and verified in the range of pH 1.0–14.0. The pH value was calculated as the average of three successive measurements. The pH meter was calibrated and verified with standard solutions, after each set of measurements. The values obtained for API-CS-oxCS/oxHA were compared with the corresponding CS-oxCS/oxHA hydrogels.

##### Rheological Behavior

The rheological measurements were performed using an Anton Paar MCR 302 rheometer (Anton Parr, Graz, Austria). To keep the working temperature constant, a Peltier system was used, with direct control over the sample temperature. The samples (CS-oxCS/oxHA and API-CS-oxCS/oxHA) were prepared 24 h before and kept at 2–8 °C. Two hours before starting the experiment, hydrogels were kept at room temperature (23 ± 2 °C) and then placed directly on a plate-plate system with striations (diameter 35 mm) to make the measurements. A solvent trap was used to avoid sample drying during the measurements. Data were interpreted with RheoCompasssoftware version V1.25.373. The working temperature was 32 °C and the measuring distance (gap) was settled at 0.5 mm for all samples.

##### Amplitude Sweep Test

This test is used to determine the limit of the linear viscoelastic range (LVE), the maximum limit of deformation tolerated by the sample without the internal structure being destroyed and to characterize the microstructure of the semisolids materials [63]. During the measurements, the amplitude is ramped (with controlled shear deformation) while the frequency is maintained constant. The set parameters are presented in Table 3. During the amplitude sweep test, the LVE (straight line on the diagram), accumulation and loss modulus (G′ and G″), and yield point were determined for each sample.

##### Thixotropic Test

The test consists in three stages: in the first one, very small deformation forces are applied (within the LVE range determined in the amplitudes sweep test) to simulate the hydrogel behavior at resting conditions. In the second step, the sample is subjected to high shear—very high deformation forces (outside the LVE) in order to simulate the breakdown of the sample. In the last stage, the minimum shear is returned with very small deformation forces to simulate the recovery of the structure [71,72]. At the end of the test, the samples recover their structure in variable percentages depending on the intrinsic physical properties. The set parameters are presented in Table 4.

The thixotropic behavior was quantified by loss factor or damping factor, tan δ, on each step of the test, which expresses the ratio between the loss (G″) and accumulation (G′) modules at rest conditions, on deformation, and after the sample recovered their internal structure, using the following formula:Tan δ = G″/G′(2)

The recovery rate was determined with respect to the G′ restoration after the sample was exposed to 700% shear strain.

#### 2.2.3. Biological Evaluation Using In Vitro Assays

##### Cell Viability Assay

The cytotoxicity degree of CS-oxCS/oxHA and API-CS-oxCS/oxHA was assessed by an MTS assay using the CellTiter 96^®^AQueous One Solution Cell Proliferation Assay (Promega, Madison, WI USA), according to the manufacturers’ instructions and extract dilution method [73]). For this purpose, the hydrogel samples’ extracts were done in a complete cell culture medium at 1% (*v*/*v*), for 24 h at 37 °C. Human gingival fibroblasts were seeded at 0.4 × 10^5^ cells/mL into 96-well tissue culture-treated plates in an MEMα medium with 10% fetal bovine serum and 1% penicillin–streptomycin–amphotericin B mixture. The next day, the cells were incubated in triplicate for 72 h with different concentrations (*v*/*v*) of hydrogel samples’ extracts (0.25%, 0.5%, 0.75%, and 1%). MTS reagent was added and absorbance readings were done 3 h later at 490 nm on a FLUOstar^®^ Omega microplate reader (BMG LABTECH, Ortenberg, Germany). Cell viability was expressed as a percentage of the control cells’ viability (means ± SD).

##### Antimicrobial Assay

The antimicrobial screening of the API-CS-oxCS/oxHA hydrogels was determined using a disk diffusion assay [74,75] against different reference strains, namely *Staphylococcus aureus* ATCC 25923, *Escherichia coli* ATCC25922, and *Candida albicans* ATCC90028. All microorganisms were stored at −80 °C in 20% glycerol. The bacterial strains were refreshed on nutrient agar, and the yeast strain was refreshed on Sabouraud dextrose agar at 37 °C. Microbial suspensions were prepared with these cultures in a sterile solution to obtain turbidity that is optically comparable to that of 0.5 McFarland standards. Volumes of 0.1 mL from each inoculum were spread on the Petri dishes. The sterilized paper disks (6 mm) were placed on the plates and aliquots (100 μL) of the samples were added. To evaluate the antimicrobial effects, the growth inhibition was measured under standard conditions after 24 h of incubation at 37 °C. All tests were carried out in triplicate. After incubation, the samples were visualized with SCAN1200^®^, version 8.6.10.0 (Interscience, Saint Nom la Brétèche, France), and the results were analyzed using XLSTAT Ecology version 2019.4.1 software and expressed as the mean ± SD [76].

## 3. Results and Discussions

### 3.1. Physicochemical Characterization of CS-oxCS/oxHA and API-CS-oxCS/oxHA Hydrogels

#### 3.1.1. Macroscopic and Microscopic Features

CS-oxCS/oxHA hydrogels are transparent or semitransparent and incorporate a large amount of air by stirring. The hydrogels have a yellow color that intensifies with the increase in the amount of crosslinking agent. The consistency of CS-oxHA hydrogels increases as the concentration of oxHA increases, while for CS-oxCS, an opposite effect was observed.

The macroscopic aspect of API-CS-oxCS/oxHA based hydrogels is dependent on API embedded in a 3D polymer matrix, slightly white for Ala-CS-oxCS/oxHA, white to slightly yellow for FA-CS-oxCS/oxHA, and yellow for CoQ10-CS-oxCS/oxHA. After preparation, the API-CS-based hydrogels were stored at 2–8 °C.

In polarized light, CS-oxCS/oxHA hydrogels show a homogeneous aspect (Figure 2). In the case of CS-oxHA hydrogels, the microscopic analysis suggests a “gel-like” behavior, while in the case of CS-oxCS hydrogels, a “liquid-like” behavior was observed. In similar conditions, the API-CS-based hydrogels show a quasihomogeneous aspect with APIs uniformly dispersed (appearing as sparkling dots) in a 3D polymer matrix (Figure 3).

#### 3.1.2. Scanning Electron Microscopy (SEM)

The SEM images of the CS-oxCS/oxHA hydrogels revealed a 3D structure, dependent on the CS:oxCS/oxHA ratio. Interestingly, in the case of CS-oxCS hydrogels, the hydrogel with the highest content of crosslinker (CS1.0-oxCH2.0) displayed the least organized microstructure with few large pores (>600 µm). The CS1.5-oxCS1.5 and CS2.0-oxCS1.0 hydrogels show a better organized morphological aspect, however, the micropores are not very well shaped. CS-oxHA hydrogels show a better-defined matrix as the amount of crosslinker increases (Figure 4).

As expected, CS1.0-oxHAox2.0, containing 2% oxHA, shows the quasiuniform distribution of 100–300 µm pores, while CS1.5-oxHA1.5 (1.5% oxHA) and CS2.0-oxHA1.0 (1.0% oxHA) revealed a less uniform defined pores. This is explained by a reduced number of carbonyl groups available for Schiff base imino interactions when the concentration of oxHA decreases.

The morphology of APIs-CS-based hydrogels is similar to corresponding CS-oxCS/oxHA with relatively a uniform distribution of pores and homogenous dispersion of APIs (Figure 5).

#### 3.1.3. FT-IR Spectroscopy

The crosslinking of CS-oxCS/oxHA hydrogels through the formation of dynamic Schiff base-type covalent bonds between the carbonyl group of oxidized polymers (oxCS, oxHA) and the free amine group of CS, was highlighted by identifying the specific peak of the vibration of the C=N bond at 1651 cm^−1^ [77]. In addition, the decrease in the peak intensity of the -C=O (1732 cm^−1^) and -NH_2_ (2924–2854 cm^−1^) groups confirms their interaction. A new peak was observed at 1720 cm^−1^ which is characteristic of the vibrations generated by the –COOH group from the lactic acid, used as a solvent (Figure 6) [78].

On the FA-CS-oxCS/oxHA spectra (Figure 7a,b), more broad and intense peaks were observed, compared with CS-oxCS/oxHA spectra. These differences are explained by the addition of the stretching vibrations and overlapping of signals of functional groups as O-H (3550–3200 cm^−1^), C=O from COOH (around 1750 cm^−1)^, double bonds C=C (1620–1560 cm^−1^), and C-O bonds (around 1100 cm^−1^) from both the FA and polysaccharide backbone of CS, oxCS, and oxHA [79].

Ala-CS-oxCS/oxHA shows also a broad peak in the 3300–2854 cm^−1^ range as a result of the overlapping of the N-H and C-H signals from the Ala structure and the CS-oxCS/oxHA hydrogels matrix (Figure 7c,d). Moreover, an intense peak at 1180–1000 cm^−1^ resulted from the overlapping of the C-O stretching of polymers with the C-N vibrations of Ala [65].

In the CoQ10-oxCS/oxHA spectra (Figure 7e,f) there appeared a new peak at the 2980–2930 cm^−1^ range which may be assigned to the alkenil radical in the CoQ10 structure. More intense peaks at 1730 cm^−1^ and between 2000–1180 cm^−1^ are also highlighted due the presence of the carbonyl group and –C-O-C- of CoQ10, respectively [67].

#### 3.1.4. Swelling Degree

SD is an important parameter of hydrogels used in the treatment of wounds. The penetration of liquids into the pores of the 3D polymer matrix gives hydrogels the optimal hydration property necessary to heal a dry or wet wound [80,81].

For all samples, around 70% from the entire value of SD was recorded after 1 h at the start of the experiment (Figure 8). The maximum value of SD was achieved after 3 h for CS-oxHA, and after 4 h for CS-oxCS, respectively (Figure 8a). Although the SD for CS-oxCS was higher (up to 568%) than CS-oxHA (up to 440%), the hydration rate was similar for both types of hydrogels.

It was also noted that for CS-oxHA, the SD decreases proportionally with the decrease in the crosslinking agent concentration. So, CS1.0-oxHA2.0 recorded an SD up to 440%, while for CS1.5-oxHA1.5 and CS2.0-oxHA1.0, the values recorded were 282% and only 86%, respectively. This may be explained by different morphologies, as observed in the SEM images. CS1.0-oxHA2.0 has more defined pores that make water uptake more efficient. For CS-oxCS, an opposite effect was observed, the SD increasing with the decreasing of the oxCS concentration in the polymer matrix.

The SD of API-CS-oxCS/oxHA (Figure 8b) is not significantly different from the values recorded for CS-oxCS/oxHA although a decrease of 10–25% was observed at each time. This could be explained by the different interactions between APIs (Ala, FA, or CoQ10) and the free hydroxyl, amino, or carboxyl groups in the hydrogel’s matrix. The SD seems to be not influenced by the type of APIs embedded in the 3D polymer matrix.

A good SD is mandatory for a wound healing dressing, as the proper moisture content is decisive for each phase of wound healing. Epithelial migration, angiogenesis, collagen synthesis, and autolytic debridement occur in optimal moisture content [22,82]. Also, the pain perception and scar surface are reduced in optimally hydrated wounds [83]. After 6 h from the start of the experiment, the API-CS-oxCS/oxHA hydrogels have an SD of more than 400% which make them perfect to absorb the exudate in the damaged tissue, as well as a perfect medium for different kinetic release of therapeutic agents to wound sites.

#### 3.1.5. pH Value

A healthy skin has a slightly acidic pH in the range of 4.5–6.5, which inhibits pathogenic microbial development while the beneficial microbial flora is favored (the so-called “acid mantle of the skin” is formed) [84]. In the chronic wounds exudate, the pH values are shifted towards the alkaline values (6.9–8.9) and start to decrease once the signs of healing are visible [85]. pH values higher than 6.5 have been shown to increase the development of *Staphylococcus aureus*, *Staphilococcus epidermides*, *Pseudomonas aeruginosa*, *Klebsiella* spp., and other species commonly found in infected wounds. Secondary compounds resulting from microbial metabolism, such as ammonia, interfere with optimal oxygenation of the injured tissue and favor its necrosis, prolonging the healing process [86,87]. Also, bacterial metabolites contribute to maintaining the pH in the alkaline range, the formation of the biofilm, and, finally, the disruption in the physiological process of recovering the integrity and functionality of the injured tissue [85]. For these reasons, in wound healing, the pH of topical products is preferred to be in the physiological range.

The pH values recorded for CS-oxCS/oxHA and API-CS-oxCS/oxHA are in the 5.77–6.012 (Table 5), which means they are within the physiological range of healthy skin and so support their wound application. It seems that the concentration and type of the crosslinking agent do not influence the pH value of hydrogels. This is because, in each hydrogel, there is the same concentration of polymer (CS and oxCS/oxHA) and of lactic acid (0.7% *w*/*w*). Moreover, the recorded pH values ensure the stability of the APIs embedded in the 3D polymer matrix. It was reported that Ala is stable at pH 4–9 while hydrolytic decomposition occurs with strong acids and bases, FA is stable around pH 5–6.2, and CoQ10 is stable under pH 6 [62,66,69].

#### 3.1.6. Rheological Behavior

##### Amplitude Sweep Test

Oscillatory rheological measurements are used to obtain information on the stability of multiphase systems. The results are presented as a diagram with shear strain plotted on the *x*-axis and accumulation modulus G′ and loss modulus G″ plotted on the *y*-axis with both axes on a logarithmic scale [88].

The G′ provides information about the amount of internal deformation energy stored in the structure, while G″ characterizes the deformation energy lost from the system during exposure of the sample to shear forces [89]. The yield point (τ) or yield stress is the value of the shear stress at the limit of the LVE region. Larger yield stress may indicate a more stable structure and better sedimentation stability over time [90]. The diagrams for CS-oxCH/oxHA and API-CS-oxCS/oxHA are presented in Figure 9.

The amplitude sweep test gives information on the viscoelastic behavior of hydrogels. So, if the accumulation modulus (G′) is higher than the loss modulus (G″), the hydrogel exhibits a solidlike behavior while if G″ exceeds the G′, it behaves like a liquid [91,92]. Based on recorded results (Table 6), it appreciates that CS-oxCS hydrogels present a structured liquid behavior, having G” higher than G′.

It was noted that G″ increases with the CS:oxCS ratio, so the CS2.0-oxCS1.0 (CS:oxCS = 2:1) exhibits the higher internal energy lost. In turn, the CS1.0-oxCS2.0 (CS:oxCS = 1:2) has the lowest internal energy lost, however, microcracks were observed on the G′ evolution in the LVE range, indicating a gradual breakdown of the microstructure. The yield points and LVE ranges for all of the CS-oxCS hydrogels are similar as a result of the liquid-like behavior.

On the other hand, CS-oxHA hydrogels have a gel structure, since the G′ modulus is much higher than the G” modulus. The best gel structure has CS1.5-oxHA1.5 in which the CS:oxHA is 1:1. For this hydrogel, the G′ modulus is approximately 2.2 times higher than CS1.0-oxHA2.0 (CS:oxHA = 1:2) and around 12 times higher than CS2.0-oxHA1.0 (CS:oxHA = 2:1), respectively. This suggests a stronger reticulation in the 3D matrix of CS1.5-oxHA1.5 and a higher stiffness of the hydrogel. The highest yield point (452.440 Pa) of CS1.5-oxHA1.5 indicates a superior resistance to shear stress, though the LVE is the smallest (77%), which means the hydrogel has the lowest stability over time.

In the case of API-CS-oxHA hydrogels, it was observed that the LVE range increases, which suggests an improvement in the stability over time. This could be explained by the new molecular interactions (e.g., hydrogen bonds) which may have occurred between free hydroxyl, amino or carboxyl groups, and reactive groups of APIs (as primary or secondary amines in Ala and carboxyl or hydroxyl in FA). In the case of API-CS-oxCS hydrogels, the range of LVE remains similar to that of the CS-oxCS hydrogels.

The G′ modules decreased in all API-CS-oxCS/oxHA, compared to CS-oxCS/oxHA, as a result of the internal friction between the API particles during exposure to shear forces.

The G″ increases in API-CS-oxHA (with solid-like behavior) while it decreases in API-CS-oxCS (liquidlike behavior) hydrogels. In the hydrogels with structured liquid rheology (API-CS-oxCS), the APIs particles will slide over each other easily due to the fluid consistency of the hydrogel base, and the energy losses under shear stress are reduced.

In the API-CS-oxHA hydrogels, the APIs particles will face the strength of the 3D matrix when exposed to various shear rates and the internal structures will lose more energy. For the same reason, the yield points remain similar to the API-CS-oxCS. In API-CS-oxHA, the yield points’ values decrease 5–12 times compared to the CS-oxHA hydrogels.

##### Thixotropic Test

The formulations designed for wound management may be exposed to various mechanical forces when an application on the damaged tissue is performed (e.g., tube extrusion, pumping, and spreading). If the product has the ability of self-healing, its structure will recover after the stress and the benefits of the healing process will be maximized [93].

The thixotropy provides information on the behavior of the analyzed samples when they were subjected to mechanical stress, and imposes the analysis of the sample before application, during the application, and after the removal of shear forces [94,95].

By applying a high shear force, over the LVE range, the hydrogel network was broken and the value of the G′ modulus suddenly decreased. In the case of hydrogels with self-healing capacity, G′ returned to the initial values (total restoration of the network) or close to the initial values (partial restoration of the 3D structure), when the shearing force was removed [93,95].

In our experiments, the LVE for all samples, CS-oxCA/oxHA and API-CS-oxCS/oxHA were under 102%. So, to assess the self-healing ability, we exposed the sample to 700% shear strain. The diagrams and the results are presented in Figure 10 and Table 7, respectively.

All analyzed samples show thixotropic behavior and recover their mechanical properties when the applied deformation forces are removed. These results support the self-healing capacity of the hydrogels crosslinked through dynamic Schiff base-type imine bonds and their potential for wound healing. The behavior is explained by the loosening of the dynamic bonds under the application of shear. When the mechanical stress is removed, the covalent interactions are restored and the gel regains its 3D structure.

For CS-oxCS/oxHA, the breaking of the internal structure occurred immediately when the shear rate was increased from ɣ = 0.1% to ɣ = 700%. The loss factor, tan δ, multiplied from 1.7 to 285 times in the deformation stage compared to the rest conditions, depending on the rheological behavior (“liquid or solid like”) and crosslinker (oxCS/oxHA) concentrations. The CS-oxCS hydrogels had recovered almost completely (>95%) their internal structure after the high shear forces were removed and the hydrogels were able to relax under a very low shear rate. Although the recovery was not significantly influenced by the amount of crosslinking agent, rate recovery was higher as the ratio between CS-oxCS increased, CS1.0-oxCS2.0 (1:2) recovered 95.338%, while CS1.5-oxCS1.5 (1:1) and CS2.0-oxCS1.0 (2:1) recovered more than 99% from their structure. As the difference in recovery for the last two hydrogels was around 0.2% we could conclude that after the optimal ratio between CS and oxCS is achieved (1:1), the increase in the crosslinker agent does not provide significant advantages in the self-healing ability of the hydrogels.

The tan δ value in the deformation stage varied at an inverse proportionality with the CS:oxCS ratio, increasing 285 times for CS1.0-oxCS2.0, almost 104 times for CS1.5-oxCS1.5, and only 9.54 times for CS2.0-oxCS1.0. This is explained by the presence of a higher amount of crosslinker agent which enriches the hydrogels with numerous dynamic Schiff bonds. These interactions are broken under mechanical stress, however, in being dynamic, they were restored when the stress was removed. So, the lower the loss factor in the second step of the thixotropic test, the recovery rate increased for the CS-oxCS hydrogels.

In the case of the CS-oxHA hydrogels, it was noted that there was an important decrease in their recovery as the CS:oxHA ratio was increasing (that means the decrease in the oxHA concentration). CS1.0-oxHA2.0 recovered 99.618% from its structure while CS1.5-oxHA1.5 recovered 36.456% and CS2.0-oxHA1.0 regained only 13.691% from their structures. So, in cases of these hydrogels with “solid-like” rheological behavior, the concentration of crosslinker (oxHA) plays a crucial role in the self-healing ability.

In the deformation step of CS-oxHA hydrogels, the tan δ had the same evolution with CS-oxCS, decreasing with the crosslinker concentration. Although the “gel-like” hydrogel was more stable under mechanical stress, tan δ increased only 34 times for CS1.0-oxHA2.0 and around 12 times for CS1.5-oxHA1.5 and 8 times for CS2.0-oxHA1.0, which means the energy losses were higher (G″ is higher than G′) under stress conditions in hydrogels with an increased concentration of crosslinker since there are more dynamic bonds subjected to breakage.

The CS-oxCS/oxHA hydrogels (CS1.5-oxCS1.5 and CS1.0-oxHA2.0) selected for embedding the APIs (FA, Ala, and CoQ10) have demonstrated more than 99% self-healing ability.

For the API-CS/oxCS hydrogels, a higher sensitivity at the application of deformation forces was observed and the recovery was down to around 30%. In addition to these, microcracks in the internal structure were observed at very low shear strains (0.1%) as a result of the frictions occurring between the API particles. The recovery rate was not influenced by the nature of the APIs. All three API-CS-oxCS hydrogels (FA-CS-oxCA, Ala-CS-oxCS, and CoQ10-CS-oxCS) recovered more than 70% after exposure to extremely high shear forces. However, the loss factor (tan δ) in the deformation stage was around two times higher for CoQ10-CS-oxCS compared to FA-CS-oxCA and Ala-CS-oxCS. This could be related to the particle size distribution of CoQ10 or different kinds of interactions between its molecule and hydrogel matrix.

The internal structure was broken immediately after 700% shear strain was applied to the API-CS-oxHA hydrogels. More differences in the recovery rate for these hydrogels were noted, in reference to API-CS-oxCS. Thus, Ala-CS-oxCS recovered 81.755% from its structure while FA-CS-oxCA and CoQ10-CS-oxCS have a restoration of around 70%.

The loss factor (tan δ) was almost 10 times higher in the CoQ10-CS-oxHA hydrogels than the other two (Ala-CS-oxHA and FA-CS-oxHA), being similar to CoQ10-CS-oxCS. The sensitivity to the application of very low deformation forces was not observed, as no microcracks occurred.

### 3.2. Biological Evaluation Using In Vitro Assays

#### 3.2.1. Cell Viability Assay

In vitro biocompatibility of CS-oxCS/oxHA and API-CS-oxCS/oxHA was assessed by the extract dilution method and MTS assay after 72 h incubation. All samples were biocompatible (>70% cell viability) at the tested concentrations (0.25%, 0.5%, 0.75%, and 1%) except FA-CS-oxHA which induced a dose-dependent decrease in cell viability and determined 60% cell vialility at 1% (*v*/*v*) concentration (Figure 11).

#### 3.2.2. Antimicrobial Assay

The API-CS-oxCS/oxHA hydrogels did not present antimicrobial activity against *Escherichia coli* as a Gram-negative bacterial strain and against *Candida albicans* as a yeast strain. Referring to the activity on the Gram-positive bacterial strain *Staphylococus aureus,* it was noted that the hydrogels containing FA (FA-CS-oxCS and FA-CS-oxHA) were very efficient, with up to 36 mm of inhibition zone (36.52 ± 0.28 mm for FA-CS-oxCS and 36.04 ± 0.49 mm for FA-CS-oxHA) (Figure 12). This effect was a bit more intense than that of the FA (31.45 ± 0.35 mm) and was frequently used to treat skin infections as well as chronic bone and joint infections [63]. This difference could be due to CS, which is known to have both antibacterial and fungicidal activities against different microorganisms [96,97,98].

## 4. Conclusions

The aim of our study was to design and characterize new smart self-healing and self-adapting CS-base hydrogels for wound care management using oxCS and oxHA as nontoxic crosslinkers agents. Different ratios between CS and oxidized polymers (CS:oxCS/oxHA) were used in order to study the physicochemical characteristics of hydrogels, according to their 3D structure, based on Schiff base-type covalent bonds. The dynamic imino bonds confer hydrogels self-healing ability, which means that the bonds could break and rebuild themselves in different environmental conditions. In addition, the hydrogel matrix obtained via dynamic Schiff bases is capable of adapting to the shape and depth of the damaged tissue which accelerates healing and ensures better protection. Three APIs (FA, Ala, and CoQ10) were selected to be embedded into the polymer matrix of CS-based hydrogels. The structure of the CS-oxCS/oxHA, the success of imine bond formation between free amine groups of CS and carbonyl groups of oxCS/oxHA, more exactly, was proven using spectral methods (FT-IR and ^1^H/^13^C-NMR). The inclusion of APIs into the polymer matrix was proven through FT-IR spectra. The hydrogels (CS-oxCS/oxHA and API-CS-oxCS/oxHA) were analyzed for their SD (%), and pH and the values recorded recommend them as suitable for use on damaged tissue. The SEM analysis showed differences in 3D networks depending on the type and concentration of the crosslinker. The rheological tests confirmed the capacity of all hydrogels to restore their internal structural organization after exposure to high shear forces which endowed them with the self-healing ability and high mechanical stability to various stressors. The embedding of APIs into the hydrogel matrix does not interfere with their mechanical performance which makes these smart hydrogels promising for the management of wounds as a wide variety of drugs may be included in their matrix to modulate the targeted therapeutic outcomes. Except for FA-CS-oxHA, API-CA-oxCS/oxHA hydrogels are not cytotoxic and, moreover, FA-CS-oxHA showed important antibacterial effects on the *Staphylococcus aureus* bacterial strain. In conclusion, the results of our study confirm that smart nontoxic hydrogels could be developed using oxidized biopolymers as crosslinker agents, with real potential in managing the treatment of irregular deep shaped wounds as a result of their self-healing and self-adapting properties.

## Figures and Tables

**Figure 1 pharmaceutics-15-00975-f001:**
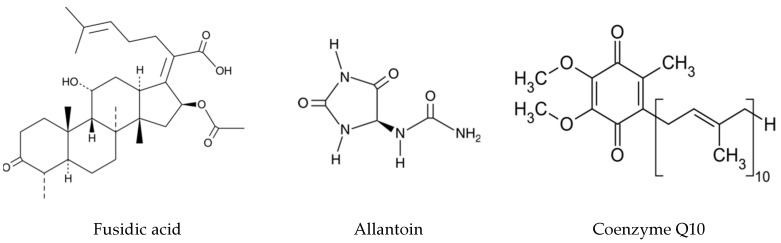
Chemical structure of APIs.

**Figure 2 pharmaceutics-15-00975-f002:**
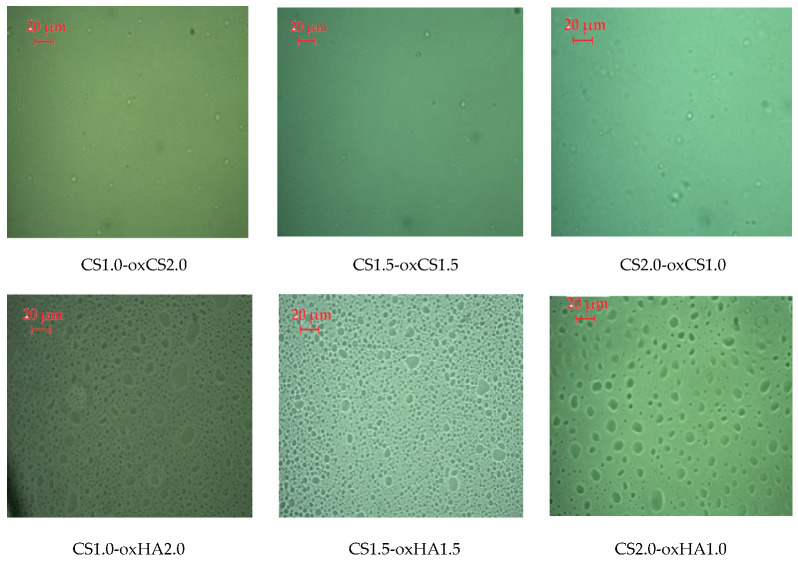
The microscopic aspect of CS-oxCS/oxHA hydrogels (40×).

**Figure 3 pharmaceutics-15-00975-f003:**
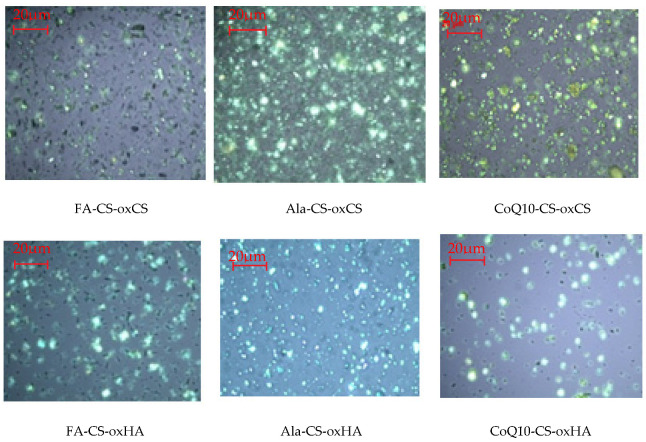
The microscopic aspect of API-CS-oxCS/oxHA hydrogels (40×).

**Figure 4 pharmaceutics-15-00975-f004:**
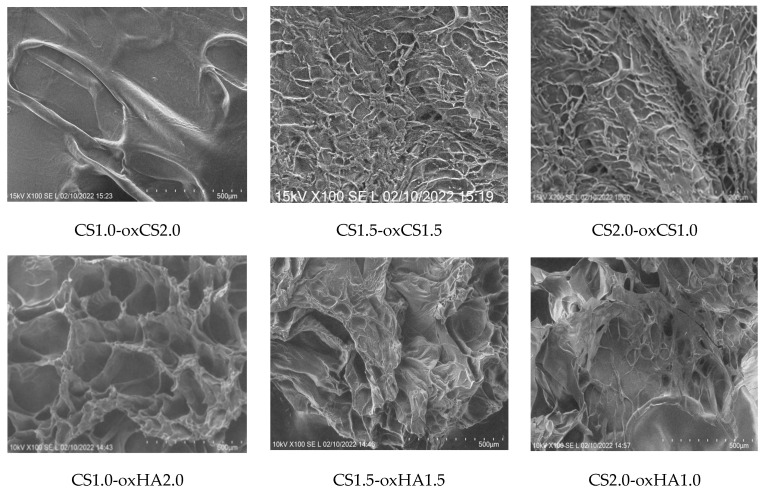
The SEM images of CS-oxCS/oxHA hydrogels.

**Figure 5 pharmaceutics-15-00975-f005:**
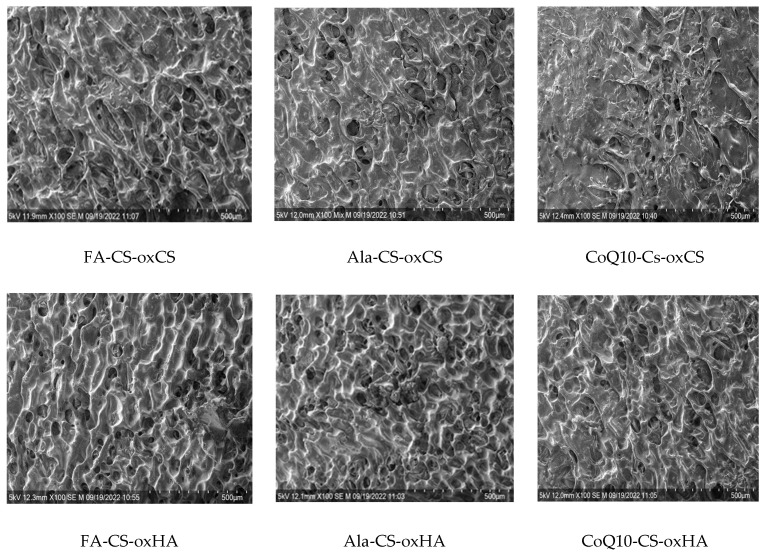
The SEM images of API-CS–oxCS/oxHA hydrogels.

**Figure 6 pharmaceutics-15-00975-f006:**
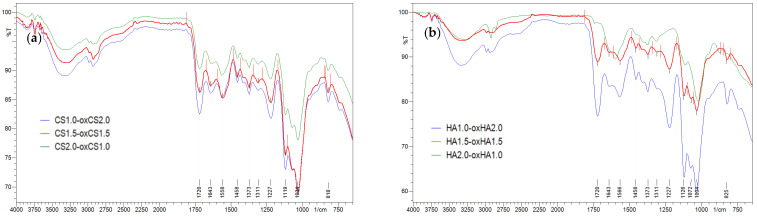
FT-IR spectra of CS-oxCS (**a**) and CS-oxHA (**b**) hydrogels.

**Figure 7 pharmaceutics-15-00975-f007:**
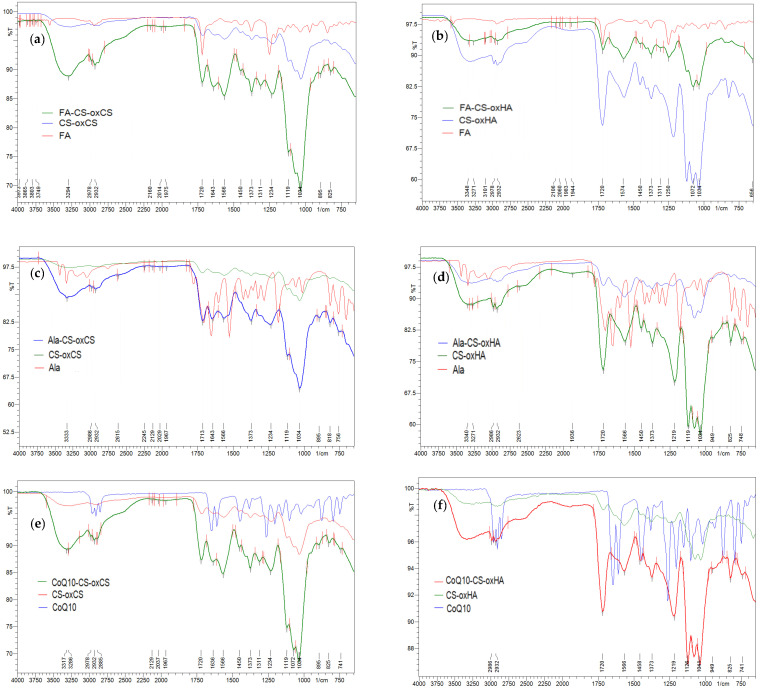
FT-IR spectra of API-CS-oxCS/oxHA hydrogels: (**a**) FA-CS-oxCS vs. CS-oxCS and FA; (**b**) FA-CS-oxHA vs. CS-oxHA and FA; (**c**) Ala-CS-oxCS vs. CS-oxCS and Ala; (**d**) Ala-CS-oxHA vs. CS-oxHA and Ala; (**e**) CoQ10-CS-oxCS vs. CS-oxCS and CoQ10: (**f**) CoQ10-CS-oxHA vs. CS-oxHA and CoQ10.

**Figure 8 pharmaceutics-15-00975-f008:**
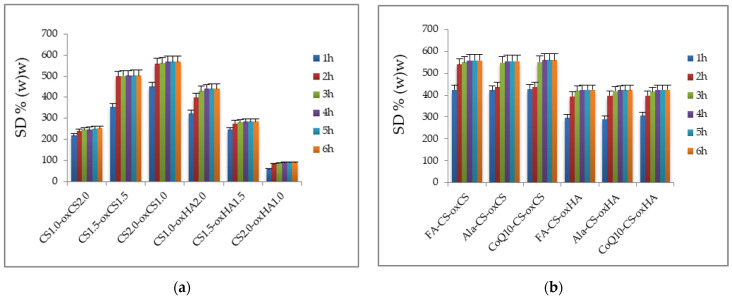
The SD (%) of CS-oxCS/oxHA (**a**) and API-CS-oxCS/oxHA (**b**) hydrogels.

**Figure 9 pharmaceutics-15-00975-f009:**
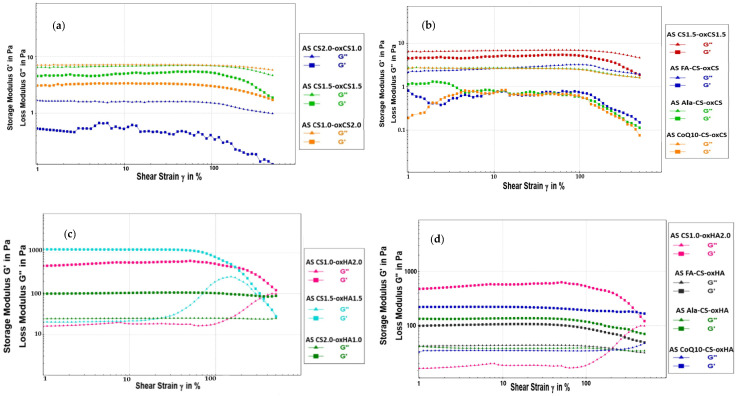
The amplitude sweep diagram for CS-oxCS (**a**), API-CS-oxCS (**b**), CS-oxHA (**c**), and API-CS-oxHA (**d**) hydrogels.

**Figure 10 pharmaceutics-15-00975-f010:**
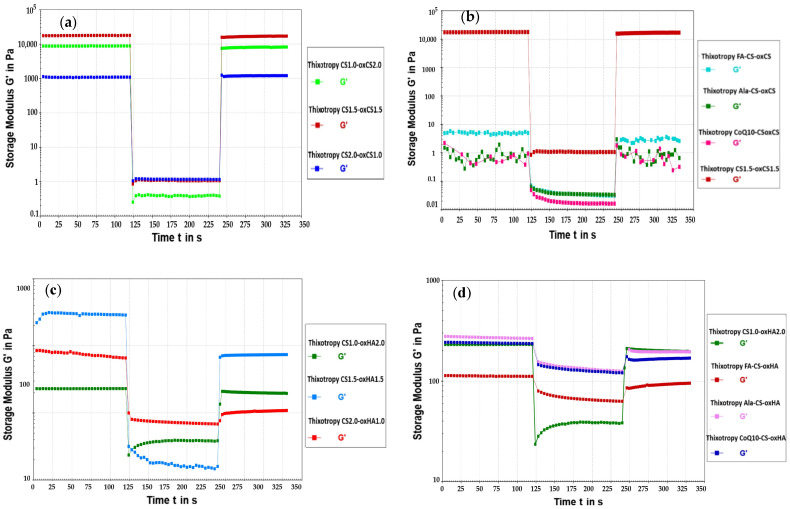
The thixotropic behavior digram of CS-oxCS (**a**), API-CS-oxCS (**b**), CS-oxHA (**c**), and API-CS-oxHA (**d**) hydrogels.

**Figure 11 pharmaceutics-15-00975-f011:**
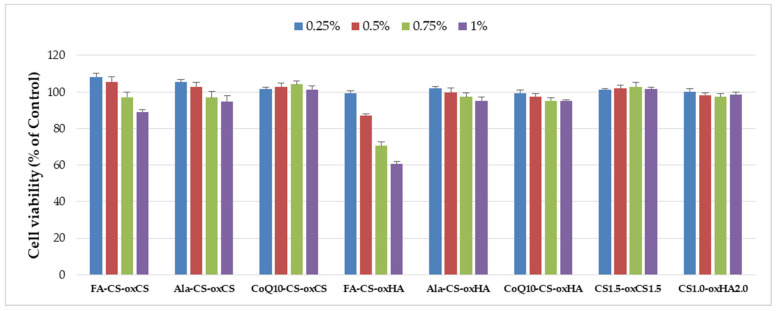
The cytotoxicity degree of CS-oxCS/oxHA and API-CS-oxCS/oxHA hydrogels.

**Figure 12 pharmaceutics-15-00975-f012:**
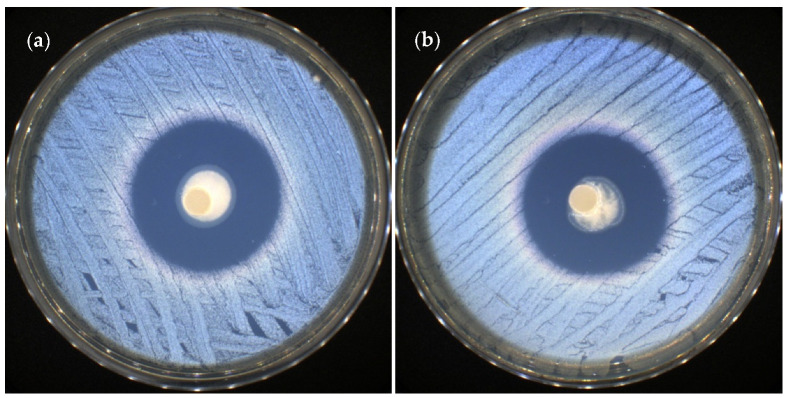
Antibacterial effect of FA-CS-oxCS (**a**) and FA-CS-oxHA (**b**) against *Staphyloccocus aureus*.

**Table 1 pharmaceutics-15-00975-t001:** The composition of CS-oxCS/oxHA hydrogels.

No.	Sample	Polymer, % (*w*/*w*)		Ratio CS:oxCS	No.	Sample	Polymer, % (*w*/*w*)		Ratio CS:oxHA
		CS	oxCS				CS	oxHA	
1	CS1.0-oxCS2.0	1.0	2.0	1:2	4	CS1.0-oxHA2.0	1.0	2.0	1:2
2	CS1.5-oxCS1.5	1.5	1.5	1:1	5	CS1.5-oxHA1.5	1.5	1.5	1:1
3	CS2.0-oxCS1.0	2.0	1.0	2:1	6	CS2.0-oxHA1.0	2.0	1.0	2:1

**Table 2 pharmaceutics-15-00975-t002:** The composition of bioactive CS-based hydrogels.

No.	Sample	API % (*w*/*w*)	CS Hydrogels
1	FA-CS-oxCS	FA, 2.0%	CS1.5-oxCS1.5
2	Ala-CS-oxCS	Ala, 1.0%	CS1.5-oxCS1.5
3	CoQ10-CS-oxCS	CoQ10, 1.0%	CS1.5-oxCS1.5
4	FA-CS-oxHA	FA, 2.0%	CS1.0-oxHA2.0
5	Ala-CS-oxHA	Ala, 1.0%	CS1.0-oxHA2.0
6	CoQ10-CS-oxHA	CoQ10, 1.0%	CS1.0-oxHA2.0

**Table 3 pharmaceutics-15-00975-t003:** The amplitude sweep test parameters.

No.	Parameter	Test Settings
1	amplitude (ɣ), %	1–500
2	angular frequency (ω), Hz	1.6
3	shear strain	oscillating
4	data points collected	100

**Table 4 pharmaceutics-15-00975-t004:** The thixotropic test parameters.

No.	Parameter	Stage I (Rest Conditions)	Stage II (Deformation)	Stage III (Recovery)
1	amplitude, ɣ %	0.1	700	0.1
2	angular frequency (ω), Hz	1.6	1.6	1.6
3	data points collected	30	30	30
4	shear strain	oscillating	oscillating	oscillating
5	test time, s	330	330	330

**Table 5 pharmaceutics-15-00975-t005:** The pH values recorded for CS-oxCS/oxHA and API-CS-oxCS/oxHA hydrogels.

No.	Sample	pH	No.	Sample	pH
1	CS1.0-oxCS2.0	5.952	7	FA-CS-oxCS	5.848
2	CS1.5-oxCS1.5	6.012	8	Ala-CS-oxCS	5.893
3	CS2.0-oxCS1.0	5.851	9	CoQ10-CS-oxCS	5.817
4	CS1.0-oxHA2.0	5.820	10	FA-CS-oxHA	5.966
5	CS1.5-oxHA1.5	5.770	11	Ala-CS-oxHA	5.939
6	CS2.0-oxHA1.0	5.805	12	CoQ10-CS-oxHA	5.968

**Table 6 pharmaceutics-15-00975-t006:** The amplitude sweep parameter values recorded for CS-oxCS/oxHA and API-CS-oxCS/oxHA hydrogels.

No.	Sample	G′ (Pa)	G″ (Pa)	LVE, ɣ (%)	Yield Point, τ (Pa)
1	CS1.0-oxCS2.0	0.206	1.629	102	0.014
2	CS1.5-oxCS1.5	4.733	6.600	101	0.002
3	CS2.0-oxCS1.0	3.238	7.086	105	0.002
4	CS1.0-oxHA2.0	507.380	16.845	83	311.010
5	CS1.5-oxHA1.5	1199.400	20.321	77	452.440
6	CS2.0-oxHA1.0	100.500	24.590	96	353.780
7	FA-CS-oxCS	0.418	2.334	100	0.015
8	Ala-CS-oxCS	1.295	2.707	98	0.003
9	CoQ10-CS-oxCS	0.423	2.673	99	0.001
10	FA-CS-oxHA	101.810	42.335	90	23.160
11	Ala-CS-oxHA	132.860	39.431	89	67.511
12	CoQ10-CS-oxHA	120.370	34.405	88	36.818

**Table 7 pharmaceutics-15-00975-t007:** The thixotropic test parameter value recorded for CS-oxCS/oxHA and API-CS-oxCS/oxHA hydrogels.

No.	Sample	Loss Factor (tan δ)			Recovery, %
		First Stage (Rest Condition)	Second Stage (Deformation)	Third Stage (Recovery)	
1	CS1.0-oxCS2.0	0.129	36.800	0.134	95.338
2	CS1.5-oxCS1.5	0.163	16.900	0.171	99.510
3	CS2.0-oxCS1.0	0.541	5.165	0.547	99.765
4	CS1.0-oxHA2.0	0.042	1.440	0.044	99.618
5	CS1.5-oxHA1.5	0.170	2.180	0.031	36.456
6	CS2.0-oxHA1.0	0.043	0.355	0.231	13.691
7	FA-CS-oxCS	0.422	28.900	1.030	73.087
8	Ala-CS-oxCS	0.410	25.200	2.730	70.556
9	CoQ10-CS-oxCS	0.854	53.900	0.918	70.943
10	FA-CS-oxHA	0.272	0.462	0.326	81.755
11	Ala-CS-oxHA	0.130	0.620	0.157	72.306
12	CoQ10-CS-oxHA	0.147	5.541	0.192	69.436

## Data Availability

The data could be requested from authors.

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
