# Peer review of "New Smart Bioactive and Biomimetic Chitosan-Based Hydrogels for Wounds Care Management"

_pharmaceutics, 2023, doi:10.3390/pharmaceutics15030975_

Round 1

Reviewer 1 Report

The article submitted for review is a great contribution to the development of the hydrogel subject related to the production of the latest generation dressing materials. The article is very well written and is a very good compendium of knowledge about hydrogels in dressings. Importantly, the authors prove in detail why each component added to the hydrogel is so important and what role it plays - the research is well thought out and not accidental. The selection of research techniques for physicochemical analysis is also very well thought out and the results are very well described. In my opinion, the only weakness of the articles are the drawings, actually all of them should be improved! Such a great article cannot contain such illegible and unfinished graphics!

Figure 1. The scheme is too trivial and not very aesthetic

Figures 5, 14-16 (FTIR) are very illegible, lack of axis captions, too small fonts and inconsistency in marking wavelengths

Figures 6 and 7 are insulting to the readers! They look like scans from an old monitor and are completely unreadable! Current free graphics programs allow you to present the results in an aesthetic and legible way!

Figures 8 and 9 are completely unnecessary - a description in the text would definitely be enough, or you can work on the aesthetics of the photo. Stains on watch glasses or photos of sample containers are not indicators of a good scientific article.

Figures 18 and 19 are completely illegible!

Author Response

Dear Reviewer,

We are very grateful for your evaluation of the quality of our manuscript and your constructive suggestions. All changes in the manuscript were added in red color and for detailed answers please see the attached document.

Best regards,

Lecturer Florentina Lupascu

Reviewer 2 Report

This manuscript describes the preparation of hydrogels based on chitosan and oxidized chitosan (oxCS) or oxidized hyaluronic acid that are designed for wound management. The physical properties of the hydrogels were well studied. This manuscript can be recommended for publication after some issues are addressed.

 1) This work was designed for wound care with the aid of release of bioactive agents (APIs). Hence, the release behaviors of the APIs (at least in vitro) from the hydrogels can be interesting.

2) Could the bioactivity of the APIs released from the hydrogels be tested?

3) Is it possible to investigate the effect of the hydrogel in wound care with an animal model?

4) Could the repeated number and error bars of the SD be supplied in Figure 16?

5) The labels in Figure 18 and 19 can hardly be indentified.

6) Could the stability (degradation) of the hydrogels be tested?

Author Response

(The authors gave the same response as above.)

Reviewer 3 Report

Please find attached the comments to the review of the paper entitled "New smart bioactive and biomimetic chitosan-based hydrogels for wounds care management".

Author Response

Dear Reviewer,

We are very grateful for your evaluation of the quality of our manuscript and your constructive suggestions. All changes in the manuscript were added in red color and for detailed answers please see the attached document.

Best regards,

Lecturer Lupascu Florentina

Reviewer 4 Report

The authors present a study focused upon new smart chitosan (CS)-based hydrogels, using oxidized chitosan (oxCS) and hyaluronic acid (oxHA) as non-toxic cross-linkers. This article is well worthy of publication because the study is a carefully done and the findings are of considerable interesting. But the paper needs significant improvement before acceptance for publication. My detailed comments are as follows:
1. Introduction needs a detailed review of previous work, for example, where are the characteristics and advantages of the materials in this paper?

2. XPS can better study the surface structure and composition of the hydrogels. It is suggested to add XPS characterization test of the hydrogels. The author can refer to these papers, J. Hazar. Mater. 410, 2021, 124539. Chem. Eng. J. 409, 2021, 128185. Journal of Materials Science & Technology 2022, 125, 59–66. Journal of Cleaner Production 360, 2022, 131948.

3. The paper needs to avoid some small mistakes, for example, the format of references needs to be unified.

Author Response

(The authors gave the same response as above.)

Reviewer 5 Report

The manuscript falls under the scope of Pharmaceutics. The authors have claimed a smart bioactive and biomimetic chitosan-based hydrogels for wound care management however none of the studies prove the smartness, bioactivity of the developed gel, and biomimetic properties. The set of studies performed does not justify the title of the manuscript. Some of the major observations are as follows;

In the macroscopic examination, the CS1.0-oxCS2.0 seems to be slightly yellow while after addition of FA the color is almost transparent white.

Line 417: How the gels are showing a liquid-like structure if they are gel?

What may be the reason for the different textures observed in the second panel of figure 10?

Figure 11 what those sparkling dots represent?

Why the addition of different drugs changed the pattern of gels as shown in figure 13?

Peaks should be indicated in the respective FTIR spectra.

It is again surprising how the SD(%) is doubling following the drug loading (Figure 16). For eg. CS1.0-oxCS2.0 the SD is coming between the 200-300 while the same is showing the SD between 500-600% following the addition of drugs. Similar observations in other case as well.

Line 321 what do authors mean by reaction by products? How the absence of these products was confirmed?

Claimed in line 405-406 is not visible in figure 8.

Data to support the claim in line 412.

I think authors should study the effect of cross-linking agents by keeping the CS constant in all the formulations.

Line 437 how the uniform distribution of the drug in the matrix can be measured by using SEM?

Minor concerns:

Line 152 dal maybe Da

Line 196 what do authors mean by a sufficient amount

Line 256 Meida may be mean

Line 397 due the

Line 414 the third gel CS2.0

Author Response

(The authors gave the same response as above.)

Reviewer 6 Report

1. Why did the authors choose to deoxidize chitosan? Would the oxidized chitosan be directly chitosan functional amine forming C-N structure?

2. Background descriptions for hydrogel wound dressings can be strengthened by citing10.1021/acsami.1c25014, 10.1016/j.cej.2022.135691 and what are the advantages of the current work compared to published articles?

3. Figures 12 and 13, How to prepare the SEM samples? For example, before dry, whether these gels are in a state of swelling equilibrium? Details should be described since the hydrogel pore size will be hugely affected by the water content.

4. In this paper, what is the release profile of the drug (Allantoin, Fusidic acid, Coenzyme Q10) chosen by the authors? Is drug release related to physical and chemical properties of gels like morphology (refering 10.1039/C6RA10716H)?

5. Figure 18, how to prepare the gel samples for rheometry? The rheometric recording conditions should be detailed because the rheometric data are affected by many parameters. For example, before the dynamic frequency sweep assay, the linear viscoelastic range of the hydrogel should be tested. I want some comments to be made to address this point.

Author Response

(The authors gave the same response as above.)

Round 2

Reviewer 1 Report

Thank you for the corrections made. The article in its current form is most suitable for publication.

Reviewer 3 Report

Dear Editors,

In my opinion the manuscript can now be accepted for publication.

Best regards,

Reviewer 5 Report

Satisfactory